behaviour/evolution/acoustics

nonverbal vocalization, communication, speech, vocal, fundamental frequency, emotion

**Author for correspondence:**
Katarzyna Pisanski
e-mail: kasiapisanski@gmail.com

†These authors contributed equally to this study.
‡Current address: Equipe Neuro-Ethologie Sensorielle, University of Lyon/Saint-Etienne, France.

# Individual differences in human voice pitch are preserved from speech to screams, roars and pain cries

Katarzyna Pisanski[1,†], Jordan Raine[2,†] and David Reby[1,2,‡]

[1]Equipe de Neuro-Ethologie Sensorielle ENES/CRNL, University of Lyon/Saint-Etienne, CNRS UMR5292, INSERM UMR_S 1028, Saint-Etienne, France
[2]Mammal Vocal Communication and Cognition Research Group, School of Psychology, University of Sussex, Brighton, UK

KP, 0000-0003-0992-2477; JR, 0000-0003-0504-6019; DR, 0000-0001-9261-1711

Fundamental frequency ($F0$, perceived as voice pitch) predicts sex and age, hormonal status, mating success and a range of social traits, and thus functions as an important biosocial marker in modal speech. Yet, the role of $F0$ in human nonverbal vocalizations remains unclear, and given considerable variability in $F0$ across call types, it is not known whether $F0$ cues to vocalizer attributes are shared across speech and nonverbal vocalizations. Here, using a corpus of vocal sounds from 51 men and women, we examined whether individual differences in $F0$ are retained across neutral speech, valenced speech and nonverbal vocalizations (screams, roars and pain cries). Acoustic analyses revealed substantial variability in $F0$ across vocal types, with mean $F0$ increasing as much as 10-fold in screams compared to speech in the same individual. Despite these extreme pitch differences, sexual dimorphism was preserved within call types and, critically, inter-individual differences in $F0$ correlated across vocal types ($r = 0.36$–$0.80$) with stronger relationships between vocal types of the same valence (e.g. 38% of the variance in roar $F0$ was predicted by aggressive speech $F0$). Our results indicate that biologically and socially relevant indexical cues in the human voice are preserved in simulated valenced speech and vocalizations, including vocalizations characterized by extreme $F0$ modulation, suggesting that voice pitch may function as a reliable individual and biosocial marker across disparate communication contexts.

# 1. Introduction

In addition to being the carrier of language, the human voice has been shaped by selection to communicate biologically relevant traits of the vocalizer ([1] for review). Indexical cues to a vocalizer's identity, sex and age are readily transmitted by the non-linguistic properties of modal speech, particularly fundamental frequency ($F0$), perceived as voice pitch ([2] for review). Fundamental frequency in modal speech corresponds to the rate at which the vocal folds vibrate, determined by their effective mass, length and tension. In addition to fluctuating dynamically during speech production within a single vocalizer ([3,4] for reviews), $F0$ varies widely across individuals, both between the sexes (men's voices are lower-pitched than women's) and within adults of the same sex, due to a combination of endocrinological, physiological, anatomical and social factors [5].

Voice pitch plays an important role in human social interactions. In addition to reliably indicating sex and age, individual differences in the mean pitch of speech utterances facilitate speaker recognition, predict mating success and social status, and influence listeners' biosocial judgements of vocalizers ([2,6,7] for reviews). However, past work in this area has focused almost exclusively on the role of pitch in speech, largely ignoring the variable nature of $F0$ across a much wider range of human vocal sounds, and thus overlooking the complexity of the human vocal apparatus and its evolved functions.

Humans can readily modulate their voice $F0$ to express a range of emotions and motivations during speech production [3,4], and studies conducted in real-life contexts have shown that men and women alter their speech $F0$ depending on social context or to whom they are speaking (see e.g. [8,9]). Yet beyond verbal communication, humans also frequently produce non-linguistic vocalizations ranging from laughter to screams that also contain salient $F0$ cues of potential biological and social relevance [10]. Such vocalizations almost certainly emerged in the human vocal repertoire before words [3], and can exploit a broader, less constrained acoustic space than modal speech. Indeed, intelligible speech production necessitates a relatively steady $F0$ ensured by an ability to maintain constant subglottal pressure [2], as well as a dense harmonic structure (thus relatively low $F0$) to effectively encode phonetic information, including discriminable formant frequencies [11]. By contrast, human nonverbal vocalizations, which can resemble those of non-human mammals (e.g. [12]), are often characterized by extreme fluctuations in voice pitch and, like those of other mammals, their acoustic structure typically maps onto their purported biological or social function [13]. This form-function mapping across vocal types appears to function to exploit listeners' perceptual sensitivities and biases. For example, distress vocalizations (cries, screams) are typically several orders of magnitude louder and higher-pitched than modal speech, attracting attention and eliciting arousal [14]. Such distress vocalizations also exhibit different spectral profiles than those of agonistic or aggressive vocalizations (roars, grunts), which are typically characterized by a relatively low $F0$ and a high proportion of nonlinear phenomena (e.g. [15,16]).

Given this considerable variability, an important question is whether acoustic cues to vocalizer attributes are shared across call types (e.g. as observed in red deer [17]) or are call-specific (e.g. as observed in zebra finches [18]). Indeed, in humans, no previous study has examined whether individual differences in $F0$ are preserved across the broad range of human vocal sounds, from modal and emotionally valenced speech to nonverbal vocalizations. While it is possible that idiosyncratic differences in $F0$ between different vocalizers might be overridden by the more extreme $F0$ modulations that characterize agonistic and distress vocalizations, existing data on the $F0$ profiles of human grunts [15], roars [16], laughs and cries [19] indicate that such vocalizations retain a degree of sexual dimorphism, wherein men produce relatively lower-pitched vocalizations than do women. There is also preliminary evidence that within each sex, $F0$ in modal speech correlates with $F0$ in sung speech [20], and that cues to individual identity are retained in valenced human speech [21], laughter [22], cries [23], and in the screams of both humans ([24], cf. [25]) and non-human primates [26] (with the caveat that speaker recognition is substantially reduced from these vocalizations compared to modal speech among human listeners [21,22,25]). Finally, longitudinal studies have recently revealed that individual differences in $F0$ remain relatively stable across the lifespan, from infancy to childhood and throughout adulthood [27,28]. Taken together, this body of literature suggests that $F0$ may represent a reliable and surprisingly stable individual marker despite its extreme dynamicity at the within-vocalizer level (e.g. across vocal types). From an evolutionary and ethological perspective, such $F0$ stability could function to provide honest information about a vocalizer's identity and biosocial profile (e.g. dominance, masculinity) regardless of the mode of vocal production, or social context.

Here, we investigate the stability of individual differences in $F0$ across speech and nonverbal vocalizations produced by the same men and women in neutral, aggressive, fearful and pain contexts, by contrasting within-individual variation in $F0$ between and across these diverse call types.

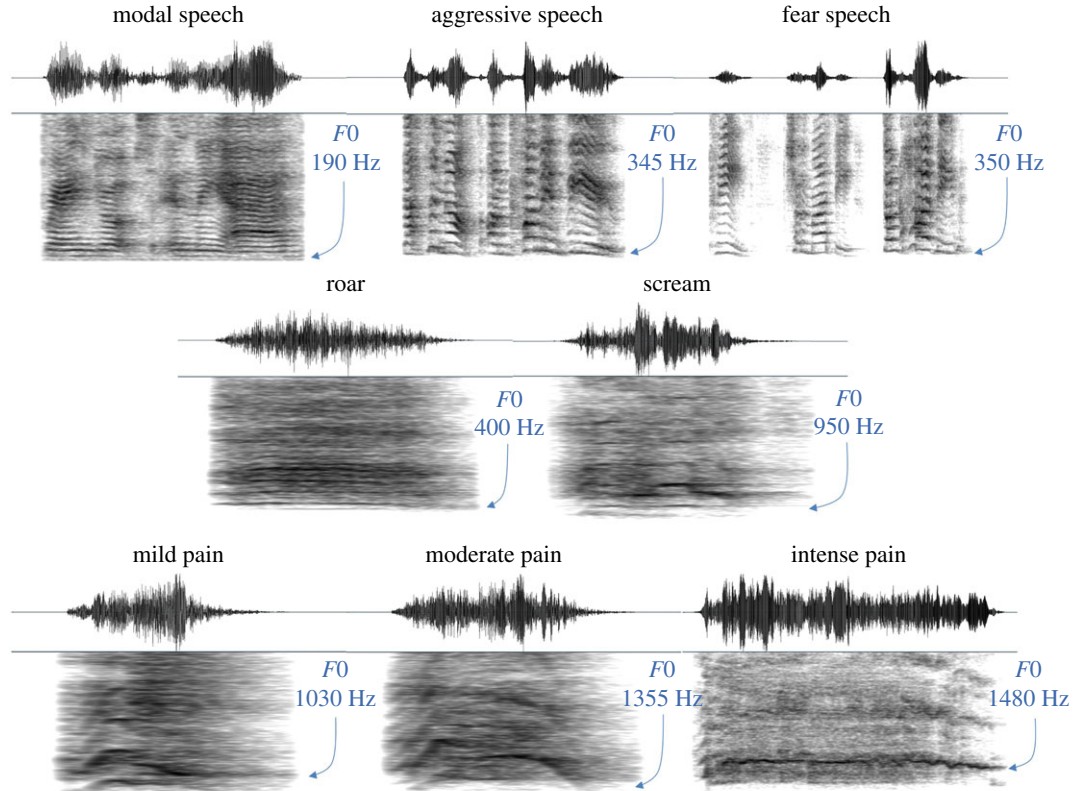

**Figure 1.** Waveforms and spectrograms representing each of eight vocal types produced by a single representative individual (female, aged 22), demonstrating the high degree of intra-individual variability in $F0$ across vocal types.

# 2. Material and methods

## 2.1. Participants

We recorded the voices of 51 drama or acting students from the Royal Central School of Speech and Drama (London, UK) and the University of Sussex (Brighton, UK). The sample included 28 men (mean age 23.6, range 18–71) and 23 women (mean age 22.8, range 18–66) who provided informed consent and received monetary compensation. No participant reported conditions that could affect their voice (e.g. cold, sore throat, chronic smoking).

## 2.2. Voice recording

Participants were audio recorded in a quiet room using a Zoom H4n microphone placed at a distance of 150 cm to prevent audio clipping. Each vocalizer produced eight simulated 'vocal types': neutral speech, aggressive speech, fearful speech, an aggressive vocalization (i.e. 'roar'), a fearful vocalization (i.e. scream), and pain vocalizations corresponding to mild, moderate and intense levels of pain (figure 1). Voice recordings were saved as WAV files at 44.1 kHz sampling frequency and 16-bit resolution, and later transferred to a laptop for acoustic analysis.

To obtain neutral modal speech, participants produced the first sentence of the Rainbow Passage [29] in a modal voice. They were then instructed to imagine themselves in various scenarios and to produce speech sentences and/or vocalizations to express their motivation and emotion in each given context. A description of the context or speech sentence was dictated by the experimenter and also displayed on a computer screen. In the aggression and fear contexts, participants imagined themselves on a battlefield, either attacking or being attacked, and produced the sentences, 'That's enough, I'm coming for you!' and, 'Please, show mercy, don't hurt me!', respectively. They were further instructed to produce a nonverbal vocalization to express that same motivation. In the pain context, participants produced vocalizations in three imagined scenarios of increasing pain intensity: mild (e.g. stubbing their toe), moderate (e.g. breaking their arm) and intense (e.g. childbirth). Full context descriptions are given in Raine *et al.* [16,30].

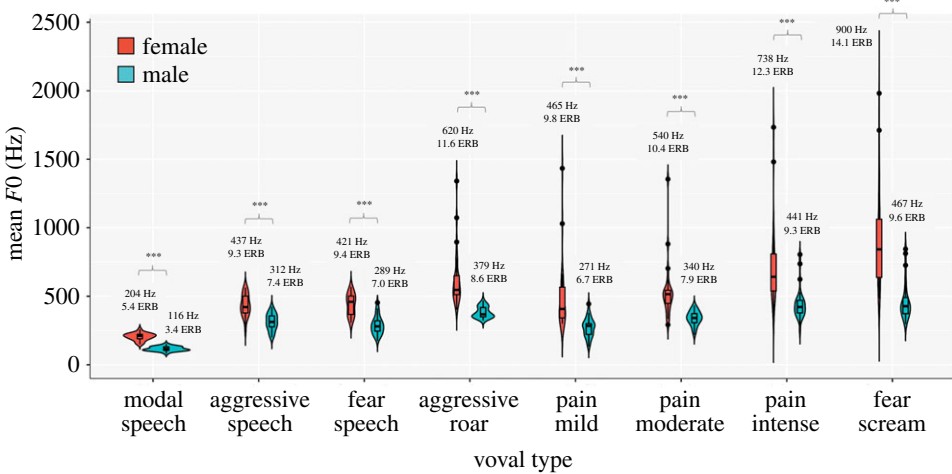

**Figure 2.** Violin plots representing the full distribution in mean $F0$ for each sex and each vocal type. Mean $F0$ values are given in both Hertz and ERBs above the violin plot for each sex and each vocal type. Significant sex differences were observed for each vocal type (***$p < 0.001$ following Šidák correction). Plots were produced in R ggplot 2 package.

## 2.3. Acoustic analysis

Acoustic editing and analysis were performed in PRAAT 5.3.62 [31]. Recordings were segregated by vocal type, resulting in 408 audio clips, and edited manually to remove silences or acute background noises. Fundamental frequency ($F0$) was measured using a custom script with a search range of 60–2000 Hz, 0.05 s window length and 0.01 time step. Extracted $F0$ contours were systematically inspected and verified, and any measurement errors (e.g. arising from octave jumps, sub-harmonics or deterministic chaos) were de-selected or corrected. This established method has been applied successfully in a number of studies to measure $F0$ both in human speech and nonverbal vocalizations characterized by extreme $F0$ values (e.g. babies cries [32], tennis grunts [15]). Measured $F0$ was converted from hertz (Hz) to equivalent rectangular bandwidths (ERBs), a quasi-logarithmic scale that controls for any discrepancy between measured $F0$ and perceived voice pitch.

## 2.4. Statistical analysis

A linear mixed model (LMM) fit by restricted maximum-likelihood estimation was first used to test for differences in $F0$ (ERB) across vocal types and between sexes. Vocal type and sex of vocalizer were entered into the omnibus model as fixed variables, and vocalizer ID was entered as a random variable with random intercept. This was followed by separate LMMs for each vocal type to more closely examine sex differences. Significant effects were examined using pairwise tests with Šidák correction for multiple comparisons.

Correlation matrices (Spearman's *rho*, $r_s$, one-tailed) were then conducted to test for positive relationships in $F0$ between all vocal types, separately for each sex and each $F0$ scale (ERB, Hz). Shapiro–Wilk tests indicated that $F0$ was not normally distributed in nonverbal vocalizations (see electronic supplementary material, table S1), hence non-parametric tests were used for regression analyses. Bootstrapping was used to compute 95% confidence intervals for each bivariate correlation. The Benjamini–Hochberg procedure ([33], where $m = 28$, $q = 0.1$) was employed to control for the inflated false discovery rate owing to multiple comparisons. Datasets and statistical scripts are provided as electronic supplementary material.

## 3. Results

### 3.1. Sexual dimorphism in $F0$ across vocal types

Figure 2 illustrates distributions in mean $F0$ across the eight vocal types for each sex. For illustrative purposes $F0$ values are plotted in Hertz along the $y$-axis and means are given in both Hertz and ERBs above each violin plot.

The omnibus LMM was significant (intercept: $F_{1,49} = 4949.6$, $p < 0.001$) and showed significant effects of vocalizer sex ($F_{1,49} = 123.1$, $p < 0.001$), vocal type ($F_{7,343} = 121.4$, $p < 0.001$) and a significant interaction between sex and vocal type ($F_{7,343} = 4.1$, $p < 0.001$) on mean $F0$.

Separate LMMs conducted for each vocal type confirmed that men's mean $F0$ was significantly lower than women's for each vocal type, including modal speech, all valenced speech and nonverbal vocalizations (all $F_{1,49} > 23.7$, all $p < 0.001$; see electronic supplementary material, table S2 for pairwise comparisons, where all $p < 0.001$ following Šidák correction). Sexual dimorphism was particularly pronounced for nonverbal vocalizations, wherein the $F0$s of women's fear screams and aggressive roars were on average 4.5 ERBs (433 Hz) and 3 ERBs (241 Hz) higher than men's, respectively. The $F0$s of women's pain cries ranged from an average of 2.6–3.2 ERBs (194–297 Hz) higher than men's (figure 2; electronic supplementary material, table S2).

## 3.2. Stability of individual differences in $F0$ across vocal types

Within each sex, significant moderate to strong positive relationships between $F0$s ($r = 0.36$–$0.80$) were also observed across various vocal types (table 1; see electronic supplementary material, table S4 for comparable results in Hz). For significant correlations, the mean $F0$ of a given vocal type explained between 13 and 64% of the variance in the mean $F0$ of another vocal type, on average, within the same sample of same-sex vocalizers (table 1).

In both sexes, $F0$ measured from aggressive speech correlated strongly with $F0$ measured from fearful speech. Moreover, both of these valenced speech types predicted the mean $F0$ of corresponding nonverbal vocalizations, where the relationships were largely valence-specific: hence, aggressive speech $F0$ predicted roar $F0$, whereas fearful speech $F0$ predicted scream $F0$. Following the same valence-specificity, roar $F0$ did not significantly predict scream $F0$ in either sex (though these weak relationships approached significance, $r = 0.26$ and $0.30$, $p < 0.10$). Within pain vocalizations, $F0$ correlated across at least two pain intensity levels in either sex, with significant relationships between mild and moderate or intense pain (table 1; electronic supplementary material, table S3).

While these results were generally consistent between the sexes, some sex differences emerged. The $F0$s of pain vocalizations correlated with roar $F0$ in men only, and with scream $F0$ in women only. Interestingly, $F0$ measured from modal speech was a relatively poor predictor of $F0$ in emotionally valenced speech or nonverbal vocalizations, particularly among women, with the exception that modal speech predicted men's fearful speech and pain vocalizations.

# 4. Discussion

We show that individual differences in human fundamental frequency ($F0$, perceived as voice pitch), previously observed in neutral speech and known to function as indexical and social signals in human conversation ([2,6,7] for reviews), are also present in valenced speech and in simulated non-linguistic vocal sounds, including those characterized by extreme $F0$ values (e.g. screams reaching nearly 2000 Hz). We further show that, despite a high degree of variability in $F0$ across eight different vocal types, between-individual differences in $F0$ are preserved across speech and vocalizations in a largely valence-specific manner. For instance, the mean $F0$ of men and women's aggressive speech sentences reliably predicted the mean $F0$ of their roars, with an analogous relationship between the $F0$s of fearful speech and screams. Individual differences in $F0$ were also preserved across pain vocalizations representing varying levels of pain intensity. All vocal types also retained sexual dimorphism and thus were significantly lower-pitched among men than women, with the most salient sex differences observed for screams, roars and pain cries.

Screams showed the most extreme $F0$ values overall (figure 1) and in turn the greatest degree of variability among vocalizers both between and within sexes. For example, the mean $F0$s of women's screams ranged from 484 to 1981 Hz, representing a difference of almost 1500 Hz (11 ERBs) between the highest- and lowest-pitched women. The fearful speech was by contrast much more constrained, with mean $F0$s ranging between 307 and 570 Hz in women (a maximum difference of 263 Hz or 3.7 ERBs between women). It is thus quite remarkable that individual differences in the $F0$s of valenced speech predicted those of such extreme vocalizations. By contrast, modal speech showed the lowest degree of $F0$ variability, with a maximum difference of only 112 Hz or 2.3 ERBs between women ($F0$ range 148–260 Hz) and 72 Hz or 1.8 ERBs between men ($F0$ range 81–153 Hz). Moreover, the $F0$ of modal speech was a poor predictor of the $F0$ of both valenced speech and nonverbal vocalizations, particularly among women for whom modal speech did not predict the $F0$ of any other vocal type. These results suggest that, in addition to valance-specificity, individual differences in $F0$ are also more strongly preserved within the broader category of emotional (compared to neutral) sounds.

**Table 1.** Correlation matrix of F0 (ERB), pairwise comparisons across all vocal types. Results for men's voices are given in the top right portion of the table (significant relationships are highlighted in blue), and results for women's voices are given in the bottom left portion of the table (significant relationships are highlighted in beige). See electronic supplementary material, table S3 for exact p-values and electronic supplementary material, table S4 for analogous comparisons on Hz scale. See electronic supplementary material, figure S1 for corresponding scatterplot matrices.

| | statistic | modal speech | aggress speech | fear speech | aggress roar | fear scream | pain mild | pain mod | pain intense |
|---|---|---|---|---|---|---|---|---|---|
| modal speech | $r_s$ | | −0.06 | 0.36* | 0.16 | 0.15 | 0.80** | 0.31 | 0.39* |
| | 95% CI | | −0.41, 0.30 | −0.07, 0.68 | −0.24, 0.50 | −0.29, 0.53 | 0.63, 0.89 | −0.12, 0.65 | 0.01, 0.69 |
| aggress speech | $r_s$ | 0.09 | | 0.55** | 0.43* | 0.26 | 0.11 | 0.08 | 0.15 |
| | 95% CI | −0.33, 0.47 | | 0.15, 0.77 | 0.04, 0.75 | −0.13, 0.53 | −0.30, 0.48 | −0.28, 0.45 | −0.22, 0.48 |
| fear speech | $r_s$ | 0.04 | 0.46* | | 0.41* | 0.54** | 0.50** | 0.12 | 0.26 |
| | 95% CI | −0.47, 0.51 | 0.09, 0.75 | | 0.03, 0.68 | 0.16, 0.77 | 0.10, 0.75 | −0.23, 0.45 | −0.12, 0.58 |
| aggress roar | $r_s$ | 0.23 | 0.62** | 0.32 | | 0.26 | 0.08 | 0.22 | 0.53** |
| | 95% CI | −0.19, 0.58 | 0.30, 0.83 | −0.06, 0.64 | | −0.14, 0.58 | −0.29, 0.45 | −0.18, 0.56 | 0.23, 0.74 |
| fear scream | $r_s$ | −0.08 | 0.30 | 0.45* | 0.30 | | 0.26 | 0.28 | 0.25 |
| | 95% CI | −0.51, 0.38 | −0.14, 0.71 | 0.10, 0.71 | −0.13, 0.63 | | −0.18, 0.61 | −0.14, 0.65 | −0.13, 0.57 |
| pain mild | $r_s$ | 0.15 | −0.17 | −0.13 | −0.01 | 0.55** | | 0.39* | 0.28 |
| | 95% CI | −0.28, 0.48 | −0.59, 0.31 | −0.56, 0.29 | −0.44, 0.47 | 0.20, 0.80 | | −0.03, 0.67 | −0.12, 0.62 |
| pain mod | $r_s$ | −0.07 | −0.12 | 0.07 | −0.17 | 0.25 | 0.43* | | 0.30 |
| | 95% CI | −0.49, 0.36 | −0.51, 0.30 | −0.40, 0.49 | −0.60, 0.30 | −0.13, 0.54 | −0.01, 0.74 | | −0.11, 0.67 |
| pain intense | $r_s$ | −0.06 | 0.18 | 0.12 | 0.03 | 0.64** | 0.44* | 0.38× | |
| | 95% CI | −0.45, 0.35 | −0.25, 0.66 | −0.36, 0.57 | −0.42, 0.51 | 0.34, 0.81 | −0.03, 0.80 | −0.05, 0.70 | |

Notes: Spearman's rho ($r_s$) correlation coefficients are followed by lower and upper 95% confidence intervals (bootstrapping, 1000 samples). Highlighted cells show significant relationships where ** $p < 0.01$, * $p < 0.05$. A single relationship (marked '×') did not pass Benjamini–Hochberg correction (unadjusted $p = 0.04$).

These results complement two longitudinal studies that recently showed that the $F0$ of babies' cries predicts the $F0$ of their speech later in childhood [28], and that individual differences in $F0$ stabilize after puberty [27], suggesting that voice pitch is determined early in life and remains fairly stable throughout adulthood [27,28]. Those findings, together with the results of the present study and other recent work showing that individuals can generally be recognized when producing emotional speech [21], laughter [22], cries [23] and screams ([24], cf. [25]), suggest that the human voice functions as a stable individual marker despite its extreme dynamicity. With regards to voice $F0$, this may be due to anatomical constraints (probably linked to individual differences in vocal fold length) that maintain the relative inter-individual differences in $F0$ across the typical ranges for each vocal type. While few studies have investigated the stability of $F0$ differences across the vocal repertoires of other species, there is evidence that cues to individual identity may also be preserved across call types in non-human mammals, including red deer [17], rhesus macaques [34] and cows [35].

The results of the present study warrant replication on a larger, more representative sample, as the current study is unpowered with the small sample size of 28 men and 23 women. As a consequence, our analyses could fail to detect statistical significance in weak and even moderate correlations ($r = 0.10$–$0.30$). This is further illustrated by the broad confidence intervals observed for most correlations (table 1). Replication studies should also include individuals without acting experience, as the potential influence of vocal training on the spectral and perceptual dimensions of speech and nonverbal vocalizations remains unclear (see e.g. [36,37]). Finally, given the influence of social and cultural factors on voice production and vocal expressions of emotion, additional research is also needed to examine the generalizability of these results to other cultures.

Extending the present line of inquiry, follow-up studies may include a broader range of speech and vocalizations (e.g. positively valenced), and could test whether modal speech $F0$, which proved to be a poor predictor in the present study, correlates with the $F0$ of less extreme and comparatively 'neutral' vocalizations such as yawns. As researchers have observed some differences between simulated and spontaneous vocal sounds (e.g. identity-related information is more reliably encoded in genuine laughter [19,22]), future studies may also compare $F0$ relationships in volitional versus spontaneous vocal types, including naturally occurring vocal displays produced in real-life social contexts.

The key implication of stable inter-individual differences in voice pitch across vocal types is that honest biosocial cues about the speaker are likely to be communicated regardless of the mode of vocal production. Thus, our results also suggest that individual differences in $F0$ may be preserved across social (and emotional) contexts, though this hypothesis should be explicitly tested. This work also contributes to a small but growing body of research examining form and function in human nonverbal vocalizations, which, despite being understudied, constitute a substantial portion of the human vocal repertoire and promise to offer new insight into the evolutionary origins of vocal communication and the evolution of speech [3,10,12].

Ethics. The study was reviewed and approved by the Sciences and Technology Cross-Schools Research Ethics Committee (C-REC) of the University of Sussex (ER/JR307/4).
Data accessibility. The datasets and tables supporting this article are available as electronic supplementary material.
Authors' contributions. D.R., J.R. and K.P. designed the investigation. J.R. collected the data. J.R. and K.P. performed acoustic analyses. D.R. and K.P. performed statistical analyses. K.P. wrote the original manuscript and created figures and tables. K.P., J.R. and D.R. edited, revised and approved the final manuscript, and agree to be accountable for the work.
Competing interests. We declare we have no competing interests.
Funding. This work was partly supported by the University of Sussex through a scholarship to J.R. Author K.P. was supported by the University of Lyon IDEXLYON project as part of the 'Programme Investissements d'Avenir' (ANR-16-IDEX-0005) to D.R., and by the European Union's Horizon 2020 research and innovation programme under the Marie Skłodowska-Curie grant agreement no. 655859.

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
