## [Reviewer comments · Royal Society Open Science]

Review History

RSOS-191642.R0 (Original submission)

Review form: Reviewer 1 (Christoph Schild)

Is the manuscript scientifically sound in its present form?

Yes

Are the interpretations and conclusions justified by the results?

Yes

Is the language acceptable?

Yes

Do you have any ethical concerns with this paper?

No

Have you any concerns about statistical analyses in this paper?

No

Recommendation?

Accept with minor revision (please list in comments)

Comments to the Author(s)

I review the manuscript "Individual differences in human voice pitch are preserved from speech to screams, roars and pain cries" (RSOS-1916421) in which the authors investigate individual differences in F0 across neutral speech, valenced speech and nonverbal vocalizations. Further, they test whether individual F0 levels are correlated across vocal types.

As already pointed out by the authors, a majority of research on F0 and its role in social interactions focuses on speech. At the same time, there are only a few studies investigating F0 in other relevant human vocalizations (e.g., roars). Given a well-chosen research design, the present study provides very relevant and important insights and will, in my opinion, make a valuable contribution to the literature. However, I have a few comments that might help to improve the manuscript:

1. **Sample Size/Participants:** The main weakness of this study is the sample size. Only 28 men and 23 women were recruited. For the correlative analyses the study has power = .95 to detect $r = .57$ and $r = .62$ (alpha = .05, one-tailed), respectively. Thus, smaller but meaningful relations (e.g., $r = .30$) are rather unlikely to be captured. Given this, I would suggest that the authors report sensitivity analyses and further discuss their findings given the relatively low power of the study. I would further suggest to report 95% confidence intervals for the correlations, as these might illustrate that even non-significant effects in this study, might actually be relevant (and should therefore be investigated in larger samples).
2. **Data & Script:** The authors should provide a short description of their variables. Currently one can guess which variable is which but this should be clarified. Further, in line with policy of this journal, I would kindly ask the authors to share their analysis script(s) to allow other researchers to reproduce the results reported in the study.

Minor comments:

1. **Figure 2:** I would suggest to choose a large font size. (I would also change the theme [e.g., "+theme_classic()"] but this is just a matter of personal taste).
2. Is there any particular reason why the acoustic analyses were run in a rather outdated version of PRAAT (version 5.3.62; released 2 January 2014)? I doubt that running the analyses in a newer version (e.g., 6.1.04 released 28 September 2019) would change the results but this might still be worth mentioning.

This was a very interesting read and I think that this study will make a very valuable contribution to the literature. The manuscript is very well written and straightforward. However, given the rather small sample, I would suggest that the authors reflect on this critically and discuss this limitation.

Review form: Reviewer 2

Is the manuscript scientifically sound in its present form?

Yes

Are the interpretations and conclusions justified by the results?

No

Is the language acceptable?

Yes

Do you have any ethical concerns with this paper?

No

Have you any concerns about statistical analyses in this paper?

No

Recommendation?

Reject

Comments to the Author(s)

Review of Royal Society Open Science ms. RSOS-191642

Pisanski et al., "Individual differences in human voice pitch are preserved from speech to screams, roars and pain cries"

General comments:

This brief and generally well-written paper continues a series from these authors in which they seek acoustic commonalities, hypothetically evolutionarily-derived, across different kinds of phonation produced by humans and other mammals. For a number of reasons I think this particular contribution overstates its case, and thus requires significant revision before it will be suitable for publication.

My first issue with the paper is its general lack of clear motivation. Why would anyone ever think that male/female differences, or differences between individuals, wouldn't be more or less preserved in these kinds of vocalizations? No hypotheses are offered about differences that are expected between different kinds of vocalization, or about anticipated differences between males and females in correlations among the different kinds of phonation. In the absence of such hypotheses, no clear explanation emerges for the patterns of results, making the paper more descriptive than experimental. Without some framework for understanding what is found, the paper loses much of its interest, at least for me.

Another way in which I think the authors oversell their case is by omitting discussion of the fact that F0 also varies within and between individuals not just for biological reasons, but also due to learned and social factors. This is particularly important here due to the use of actors from a single training tradition, rather than naturally occurring utterances or at least acted performances drawn from two cultures with different emotion display rules. This point requires some discussion and should be mentioned as a significant limitation of the study.

I also don't completely agree that the acoustic space in speech is constrained by rules of language. If this is true, then why study speech during (simulated) pain or fear or aggression, as they do here? What about singing, which is linguistic, but particularly in modern vocalization styles is quite unconstrained in terms of voice quality? At a minimum, this claim requires citation and justification. Similarly, the conclusion that human voice pitch functions as a stable individual marker due to anatomical and mechanistic constraints is simplistic, given the huge range of variability an individual can produce (as their own data show). F0 ranges almost completely overlap for males and females, for every utterance type except modal speech, further undermining this argument.

Finally, in their interpretation of the data the authors seem to imply that all statistically reliable correlation values are somehow equivalent, although there's a big difference between 13% and 64% variance accounted for. Conclusions are not equally strong in all cases here, particularly given the differences between males and females in patterns of association. Again, I'm sure there's something going on here, but it seems to me that results are subtler and weaker than the authors state.

Specific quibbles:

"Hertz" should be capitalized throughout the manuscript.

Scientific writing is precise. No perceptual assessment was undertaken, so the use of “pitch” in this paper is inappropriate. Replace with “F0” throughout.

The word “despite” at the beginning of the introduction implies that use of voice for speech is somehow in conflict with non-linguistic uses of phonation, which isn’t really true. I suggest rewriting this sentence (possibly by just deleting the first clause).

Line 145: errant apostrophe in ERBs.

Review form: Reviewer 3

Is the manuscript scientifically sound in its present form?

Yes

Are the interpretations and conclusions justified by the results?

Yes

Is the language acceptable?

Yes

Do you have any ethical concerns with this paper?

No

Have you any concerns about statistical analyses in this paper?

Yes

Recommendation?

Accept with minor revision (please list in comments)

Comments to the Author(s)

This study examined whether inter-individual differences in fundamental frequency (F0) are correlated across vocal types. The authors reported novel findings that human F0 are preserved across vocal types (valenced speech and vocalizations), suggesting that F0 might be a reliable biosocial marker across communication contexts. The research question is interesting, and findings are relevant to recent reports in literature, which examine the perceptual effect of human non-verbal vocalizations (screams, roars, etc.) that are beyond modal speech. Nevertheless, there are some concerns with the study’s sample size, and these limitations should be highlighted in discussion.

Major Concerns:

-One of the concerns with the study is its sample size ($n = 23$ for women and $n = 28$ for men). The authors, for example, concluded that “...modal speech was a poor predictor of the pitch of both valenced speech and nonverbal vocalizations (p.12, line 220).” I conducted a quick G-power post-hoc analysis with one-tailed test using women ($n = 23$), spearman’s $\rho = .09$ (between modal speech and aggressive speech), and p value of .05, which produces the statistical power 0.11. Hence, the probability of avoiding a Type II error is low.

-Some of the voice recordings were collected from drama or acting students. Livingstone et al. (2014) reported that vocal training and acting experiences influence vocalizer’s pitch and other measures. <https://www.ncbi.nlm.nih.gov/pmc/articles/PMC3945712/> Perhaps, the correlation in inter-individual differences in pitch across vocal types could be driven by the unique

characteristics of the sample. Please consider reporting the number of drama students in the sample. Are there differences in F0 variations between acting students' and University of Sussex students'?

Minor Points

-Consider running independent t-tests or 2-way mixed ANOVAs (Sex x Vocal Type), with reports on Means and SDs, to report the degree of sexual dimorphism in F0 across vocal type. Judging from the figure, it appears F0 dimorphism is greatest in aggressive roar and fear scream.

-I suggest some modifications for Figure 1. What do the numbers on box plot represent? Are they IDs of the outlier, or F0 value in ERB? Also consider changing F to Female and M to Male. If not, including notes in the figure caption would clarify the points.

-It's great to see that Spearman's rho for all analyses (rs), both significant and non-significant, were reported in Table 1. Nevertheless, plotting these rank relationships in x-y plot would provide additional information. Since Spearman analyses only consider monotonic relationships, curvilinear relationships, for example, may not be detected without visualization.

-Are there similar lines of evidence in nonhuman animals, in which inter-individual differences in fundamental frequency (F0) that are correlated across vocal types? It would be a nice discussion point.

Decision letter (RSOS-191642.R0)

04-Nov-2019

Dear Dr Pisanski,

The editors assigned to your paper ("Individual differences in human voice pitch are preserved from speech to screams, roars and pain cries") have now received comments from reviewers. We would like you to revise your paper in accordance with the referee and Associate Editor suggestions which can be found below (not including confidential reports to the Editor). Please note this decision does not guarantee eventual acceptance.

Please submit a copy of your revised paper before 27-Nov-2019. Please note that the revision deadline will expire at 00.00am on this date. If we do not hear from you within this time then it will be assumed that the paper has been withdrawn. In exceptional circumstances, extensions may be possible if agreed with the Editorial Office in advance. We do not allow multiple rounds of revision so we urge you to make every effort to fully address all of the comments at this stage. If deemed necessary by the Editors, your manuscript will be sent back to one or more of the original reviewers for assessment. If the original reviewers are not available, we may invite new reviewers.

When submitting your revised manuscript, you must respond to the comments made by the referees and upload a file "Response to Referees" in "Section 6 - File Upload". Please use this to document how you have responded to the comments, and the adjustments you have made. In

order to expedite the processing of the revised manuscript, please be as specific as possible in your response.

- Data accessibility

If you wish to submit your supporting data or code to Dryad (<http://datadryad.org/>), or modify your current submission to dryad, please use the following link:
<http://datadryad.org/submit?journalID=RSOS&manu=RSOS-191642>

- Competing interests

- Authors' contributions

- Acknowledgements

- Funding statement

Once again, thank you for submitting your manuscript to Royal Society Open Science and I look

forward to receiving your revision. If you have any questions at all, please do not hesitate to get in touch.

Kind regards,
 Andrew Dunn
 Senior Publishing Editor
 Royal Society Open Science
 openscience@royalsociety.org

on behalf of Dr César Lima (Associate Editor) and Essi Viding (Subject Editor)
 openscience@royalsociety.org

Comments to Author:

Reviewers' Comments to Author:

Reviewer: 1

Comments to the Author(s)

I review the manuscript "Individual differences in human voice pitch are preserved from speech to screams, roars and pain cries" (RSOS-1916421) in which the authors investigate individual differences in F0 across neutral speech, valenced speech and nonverbal vocalizations. Further, they test whether individual F0 levels are correlated across vocal types.

As already pointed out by the authors, a majority of research on F0 and its role in social interactions focuses on speech. At the same time, there are only a few studies investigating F0 in other relevant human vocalizations (e.g., roars). Given a well-chosen research design, the present study provides very relevant and important insights and will, in my opinion, make a valuable contribution to the literature. However, I have a few comments that might help to improve the manuscript:

1. Sample Size/Participants: The main weakness of this study is the sample size. Only 28 men and 23 women were recruited. For the correlative analyses the study has power = .95 to detect $r = .57$ and $r = .62$ ($\alpha = .05$, one-tailed), respectively. Thus, smaller but meaningful relations (e.g., $r = .30$) are rather unlikely to be captured. Given this, I would suggest that the authors report sensitivity analyses and further discuss their findings given the relatively low power of the study. I would further suggest to report 95% confidence intervals for the correlations, as these might illustrate that even non-significant effects in this study, might actually be relevant (and should therefore be investigated in larger samples).
2. Data & Script: The authors should provide a short description of their variables. Currently one can guess which variable is which but this should be clarified. Further, in line with policy of this journal, I would kindly ask the authors to share their analysis script(s) to allow other researchers to reproduce the results reported in the study.

Minor comments:

1. Figure 2: I would suggest to choose a large font size. (I would also change the theme [e.g., "+theme_classic()"] but this is just a matter of personal taste).
2. Is there any particular reason why the acoustic analyses were run in a rather outdated version of PRAAT (version 5.3.62; released 2 January 2014)? I doubt that running the analyses in a newer version (e.g., 6.1.04 released 28 September 2019) would change the results but this might still worth mentioning.

This was a very interesting read and I think that this study will make a very valuable contribution to the literature. The manuscript is very well written and straightforward. However,

given the rather small sample, I would suggest that the authors reflect on this critically and discuss this limitation.

Reviewer: 2

Comments to the Author(s)

Review of Royal Society Open Science ms. RSOS-191642

Pisanski et al., "Individual differences in human voice pitch are preserved from speech to screams, roars and pain cries"

General comments:

This brief and generally well-written paper continues a series from these authors in which they seek acoustic commonalities, hypothetically evolutionarily-derived, across different kinds of phonation produced by humans and other mammals. For a number of reasons I think this particular contribution overstates its case, and thus requires significant revision before it will be suitable for publication.

My first issue with the paper is its general lack of clear motivation. Why would anyone ever think that male/female differences, or differences between individuals, wouldn't be more or less preserved in these kinds of vocalizations? No hypotheses are offered about differences that are expected between different kinds of vocalization, or about anticipated differences between males and females in correlations among the different kinds of phonation. In the absence of such hypotheses, no clear explanation emerges for the patterns of results, making the paper more descriptive than experimental. Without some framework for understanding what is found, the paper loses much of its interest, at least for me.

Another way in which I think the authors oversell their case is by omitting discussion of the fact that F0 also varies within and between individuals not just for biological reasons, but also due to learned and social factors. This is particularly important here due to the use of actors from a single training tradition, rather than naturally occurring utterances or at least acted performances drawn from two cultures with different emotion display rules. This point requires some discussion and should be mentioned as a significant limitation of the study.

I also don't completely agree that the acoustic space in speech is constrained by rules of language. If this is true, then why study speech during (simulated) pain or fear or aggression, as they do here? What about singing, which is linguistic, but particularly in modern vocalization styles is quite unconstrained in terms of voice quality? At a minimum, this claim requires citation and justification. Similarly, the conclusion that human voice pitch functions as a stable individual marker due to anatomical and mechanistic constraints is simplistic, given the huge range of variability an individual can produce (as their own data show). F0 ranges almost completely overlap for males and females, for every utterance type except modal speech, further undermining this argument.

Finally, in their interpretation of the data the authors seem to imply that all statistically reliable correlation values are somehow equivalent, although there's a big difference between 13% and 64% variance accounted for. Conclusions are not equally strong in all cases here, particularly given the differences between males and females in patterns of association. Again, I'm sure there's something going on here, but it seems to me that results are subtler and weaker than the authors state.

Specific quibbles:

"Hertz" should be capitalized throughout the manuscript.

Scientific writing is precise. No perceptual assessment was undertaken, so the use of "pitch" in this paper is inappropriate. Replace with "F0" throughout.

The word “despite” at the beginning of the introduction implies that use of voice for speech is somehow in conflict with non-linguistic uses of phonation, which isn't really true. I suggest rewriting this sentence (possibly by just deleting the first clause).

Line 145: errant apostrophe in ERBs.

Reviewer: 3

Comments to the Author(s)

This study examined whether inter-individual differences in fundamental frequency (F0) are correlated across vocal types. The authors reported novel findings that human F0 are preserved across vocal types (valenced speech and vocalizations), suggesting that F0 might be a reliable biosocial marker across communication contexts. The research question is interesting, and findings are relevant to recent reports in literature, which examine the perceptual effect of human non-verbal vocalizations (screams, roars, etc.) that are beyond modal speech. Nevertheless, there are some concerns with the study's sample size, and these limitations should be highlighted in discussion.

Major Concerns:

-One of the concerns with the study is its sample size ($n = 23$ for women and $n = 28$ for men). The authors, for example, concluded that “...modal speech was a poor predictor of the pitch of both valenced speech and nonverbal vocalizations (p.12, line 220).” I conducted a quick G-power post-hoc analysis with one-tailed test using women ($n = 23$), spearman's $\rho = .09$ (between modal speech and aggressive speech), and p value of .05, which produces the statistical power 0.11. Hence, the probability of avoiding a Type II error is low.

-Some of the voice recordings were collected from drama or acting students. Livingstone et al. (2014) reported that vocal training and acting experiences influence vocalizer's pitch and other measures. <https://www.ncbi.nlm.nih.gov/pmc/articles/PMC3945712/> Perhaps, the correlation in inter-individual differences in pitch across vocal types could be driven by the unique characteristics of the sample. Please consider reporting the number of drama students in the sample. Are there differences in F0 variations between acting students' and University of Sussex students'?

Minor Points

-Consider running independent t-tests or 2-way mixed ANOVAs (Sex x Vocal Type), with reports on Means and SDs, to report the degree of sexual dimorphism in F0 across vocal type. Judging from the figure, it appears F0 dimorphism is greatest in aggressive roar and fear scream.

-I suggest some modifications for Figure 1. What do the numbers on box plot represent? Are they IDs of the outlier, or F0 value in ERB? Also consider changing F to Female and M to Male. If not, including notes in the figure caption would clarify the points.

-It's great to see that Spearman's ρ for all analyses (r_s), both significant and non-significant, were reported in Table 1. Nevertheless, plotting these rank relationships in x-y plot would provide additional information. Since Spearman analyses only consider monotonic relationships, curvilinear relationships, for example, may not be detected without visualization.

-Are there similar lines of evidence in nonhuman animals, in which inter-individual differences in fundamental frequency (F0) that are correlated across vocal types? It would be a nice discussion point.

Author's Response to Decision Letter for (RSOS-191642.R0)

See Appendix A.

RSOS-191642.R1 (Revision)

Review form: Reviewer 1 (Christoph Schild)

Is the manuscript scientifically sound in its present form?

Yes

Are the interpretations and conclusions justified by the results?

Yes

Is the language acceptable?

Yes

Do you have any ethical concerns with this paper?

No

Have you any concerns about statistical analyses in this paper?

No

Recommendation?

Accept with minor revision (please list in comments)

Comments to the Author(s)

I review the revised manuscript "Individual differences in human voice pitch are preserved from speech to screams, roars and pain cries" (RSOS-191642.R1).

While most comments (including those of the other reviewers) have been very well addressed, I must say that one major weakness of this study, the sample size (as pointed out by me and R3), has not been well addressed in the manuscript.

A sensitivity analysis was not provided and the current discussion of this clear limitation ("The results of the present study warrant replication on a larger, more representative sample, as the current study may be unpowered to detect statistical significance in weak correlations ($r = 0.10 - 0.30$). Nevertheless, given that these relationships explain less than 10% of the variance in shared F0 across vocal types, their ecological relevance is uncertain.") seems to miss the point. First, as pointed out in my previous review, the study is even only moderately powered for $r_s > .3$. For example, with $N = 23$ you have power $\approx .62$ ($\alpha = .05$, one tailed) to detect an effect of $r = .4$ (which is typically considered a medium to large effect size). Thus, the study is, in general, clearly underpowered. Second, even if the observed estimate is high (e.g., $r = .55$) looking at the 95% CIs tells us that this is a really imprecise estimate which might, in well-powered studies, fall far below this estimate (e.g., $r = .20$).

In conclusion, I would hope that the authors can discuss this limitation in another revision. Further, I see no reason for putting the 95% CIs in the supplementary table only. It should be possible to fit those in Table 1.

Review form: Reviewer 2

Is the manuscript scientifically sound in its present form?

Yes

Are the interpretations and conclusions justified by the results?

Yes

Is the language acceptable?

Yes

Do you have any ethical concerns with this paper?

No

Have you any concerns about statistical analyses in this paper?

No

Recommendation?

Accept as is

Comments to the Author(s)

Review of RSOS-191642-R1

Pisanski et al., Individual differences in human voice pitch are preserved from speech to screams, roars, and pain cries

The authors have done an excellent job of revising this interesting paper, and have addressed all my concerns to my satisfaction. It remains only for me to quibble about apostrophe use:

Line 164: bandwidth's bandwidths

Lines 207, 208, 209: ERB's ERBs

Lines 213, 238, 261, 266: F0's F0s

Line 252: men men's

Review form: Reviewer 3

Is the manuscript scientifically sound in its present form?

Yes

Are the interpretations and conclusions justified by the results?

Yes

Is the language acceptable?

Yes

Do you have any ethical concerns with this paper?

Yes

Have you any concerns about statistical analyses in this paper?

No

Recommendation?

Reject

Comments to the Author(s)

The authors satisfactorily addressed my concerns, including other reviewers', with the manuscript. I do not have any additional comments.

Decision letter (RSOS-191642.R1)

14-Jan-2020

Dear Dr Pisanski,

On behalf of the Editors, I am pleased to inform you that your Manuscript RSOS-191642.R1 entitled "Individual differences in human voice pitch are preserved from speech to screams, roars and pain cries" has been accepted for publication in Royal Society Open Science subject to minor revision in accordance with the referee suggestions. Please find the referees' comments at the end of this email.

The reviewers and Subject Editor have recommended publication, but also suggest some minor revisions to your manuscript. Therefore, I invite you to respond to the comments and revise your manuscript.

- Ethics statement

- Data accessibility

If you wish to submit your supporting data or code to Dryad (<http://datadryad.org/>), or modify your current submission to dryad, please use the following link:
<http://datadryad.org/submit?journalID=RSOS&manu=RSOS-191642.R1>

- Competing interests

- Authors' contributions

- Acknowledgements

- Funding statement

Because the schedule for publication is very tight, it is a condition of publication that you submit the revised version of your manuscript before 23-Jan-2020. Please note that the revision deadline will expire at 00.00am on this date. If you do not think you will be able to meet this date please let me know immediately.

Best regards,
Lianne Parkhouse
Editorial Coordinator
Royal Society Open Science

on behalf of Dr César Lima (Associate Editor) and Professor Essi Viding (Subject Editor)
openscience@royalsociety.org

Reviewer comments to Author:

Reviewer: 1

Comments to the Author(s)

I review the revised manuscript "Individual differences in human voice pitch are preserved from speech to screams, roars and pain cries" (RSOS-191642.R1).

While most comments (including those of the other reviewers) have been very well addressed, I must say that one major weakness of this study, the sample size (as pointed out by me and R3), has not been well addressed in the manuscript.

A sensitivity analysis was not provided and the current discussion of this clear limitation ("The results of the present study warrant replication on a larger, more representative sample, as the current study may be unpowered to detect statistical significance in weak correlations ($r = 0.10 - 0.30$). Nevertheless, given that these relationships explain less than 10% of the variance in shared F0 across vocal types, their ecological relevance is uncertain.") seems to miss the point. First, as pointed out in my previous review, the study is even only moderately powered for $r_s > .3$. For example, with $N = 23$ you have power = .62 ($\alpha = .05$, one tailed) to detect an effect of $r = .4$ (which is typically considered a medium to large effect size). Thus, the study is, in general, clearly underpowered. Second, even if the observed estimate is high (e.g., $r = .55$) looking at the 95% CIs tells us that this is a really imprecise estimate which might, in well-powered studies, fall far below this estimate (e.g., $r = .20$).

In conclusion, I would hope that the authors can discuss this limitation in another revision. Further, I see no reason for putting the 95% CIs in the supplementary table only. It should be possible to fit those in Table 1.

Reviewer: 2

Comments to the Author(s)

Review of RSOS-191642-R1

Pisanski et al., Individual differences in human voice pitch are preserved from speech, roars, and pain cries

The authors have done an excellent job of revising this interesting paper, and have addressed all my concerns to my satisfaction. It remains only for me to quibble about apostrophe use:

Line 164: bandwidth's □ bandwidths

Lines 207, 208, 209: ERB's □ ERBs

Lines 213, 238, 261, 266: F0's □ F0s

Line 252: men □ men's

Reviewer: 3

Comments to the Author(s)

The authors satisfactorily addressed my concerns, including other reviewers', with the manuscript. I do not have any additional comments.

Author's Response to Decision Letter for (RSOS-191642.R1)

See Appendix B.

Decision letter (RSOS-191642.R2)

21-Jan-2020

Dear Dr Pisanski,

It is a pleasure to accept your manuscript entitled "Individual differences in human voice pitch are preserved from speech to screams, roars and pain cries" in its current form for publication in Royal Society Open Science. The comments of the reviewer(s) who reviewed your manuscript are included at the foot of this letter.

on behalf of Dr César Lima (Associate Editor) and Essi Viding (Subject Editor)
openscience@royalsociety.org

Associate Editor Comments to Author (Dr César Lima):
Associate Editor: 1
Comments to the Author:
(There are no comments.)

Reviewer comments to Author:

Appendix A

Response to Reviewers

RSOS-191642-R1

Reviewer: 1

I review the manuscript “Individual differences in human voice pitch are preserved from speech to screams, roars and pain cries” (RSOS-191642I) in which the authors investigate individual differences in F0 across neutral speech, valenced speech and nonverbal vocalizations. Further, they test whether individual F0 levels are correlated across vocal types.

As already pointed out by the authors, a majority of research on F0 and its role in social interactions focuses on speech. At the same time, there are only a few studies investigating F0 in other relevant human vocalizations (e.g., roars). Given a well-chosen research design, the present study provides very relevant and important insights and will, in my opinion, make a valuable contribution to the literature. However, I have a few comments that might help to improve the manuscript.

We thank the reviewer for the positive feedback and constructive suggestions, which we address in bold and in line below.

1. Sample Size/Participants: The main weakness of this study is the sample size. Only 28 men and 23 women were recruited. For the correlative analyses the study has power = .95 to detect $r = .57$ and $r = .62$ ($\alpha = .05$, one-tailed), respectively. Thus, smaller but meaningful relations (e.g., $r = .30$) are rather unlikely to be captured. Given this, I would suggest that the authors report sensitivity analyses and further discuss their findings given the relatively low power of the study. I would further suggest to report 95% confidence intervals for the correlations, as these might illustrate that even non-significant effects in this study, might actually be relevant (and should therefore be investigated in larger samples).

We agree and have now used bootstrapping to compute 95% confidence intervals for each bivariate correlation, reported in Table S3. We have also noted the lack of power as a limitation in the Discussion while remarking on the ecological relevance of weak correlations ($r_s < 0.30$):

“The results of the present study warrant replication on a larger, more representative sample, as the current study may be unpowered to detect statistical significance in weak correlations ($r = 0.10 - 0.30$). Nevertheless, given that these relationships explain less than 10% of the variance in shared F0 across vocal types, their ecological relevance is uncertain” (Lines 290 – 294)

2. Data & Script: The authors should provide a short description of their variables. Currently one can guess which variable is which but this should be clarified. Further, in line with policy of this journal, I would kindly ask the authors to share their analysis script(s) to allow other researchers to reproduce the results reported in the study.

We have now annotated our data (ESM 2 Dataset – see coding legend) such that all variables are clearly labelled, and have included scripts for all statistical analyses as supplementary material (ESM 3).

Minor comments:

1. Figure 2: I would suggest to choose a large font size. (I would also change the theme [e.g., “+theme_classic()”] but this is just a matter of personal taste).

We have chosen a larger font size and adjusted the figure style for readability.

2. Is there any particular reason why the acoustic analyses were run in a rather outdated version of PRAAT (version 5.3.62; released 2 January 2014)? I doubt that running the analyses in a newer version (e.g., 6.1.04 released 28 September 2019) would change the results but this might still worth mentioning.

Acoustic analyses were performed by author Jordan Raine during his doctoral studies which commenced in 2014. The version of PRAAT was the most up-to-date at that time. As the Reviewer notes, there have been no substantial changes to the F0-tracking algorithm used by PRAAT in subsequent versions.

This was a very interesting read and I think that this study will make a very valuable contribution to the literature. The manuscript is very well written and straightforward. However, given the rather small sample, I would suggest that the authors reflect on this critically and discuss this limitation.

Thank you again for your constructive feedback.

Reviewer: 2

This brief and generally well-written paper continues a series from these authors in which they seek acoustic commonalities, hypothetically evolutionarily-derived, across different kinds of phonation produced by humans and other mammals. For a number of reasons I think this particular contribution overstates its case, and thus requires significant revision before it will be suitable for publication.

We thank the Reviewer for taking the time to review the paper and offering suggestions for improving its clarity and impact, particularly underscoring the implications of the results. Below we respond to each of the Reviewer’s comments in bold and in line.

My first issue with the paper is its general lack of clear motivation. Why would anyone ever think that male/female differences, or differences between individuals, wouldn’t be more or less preserved in these kinds of vocalizations? No hypotheses are offered about differences that are expected between different kinds of vocalization, or about anticipated differences between males and females in correlations among the different kinds of phonation. In the absence of such hypotheses, no clear explanation emerges for the patterns of results, making the

paper more descriptive than experimental. Without some framework for understanding what is found, the paper loses much of its interest, at least for me.

We apologise that the rationale and motivations were not effectively emphasised, and as such have expanded the abstract and introductory text and reformulated several statements to more clearly and explicitly underscore these points. The implications of our results are outlined in the Discussion.

Below we paste an excerpt but invite the Reviewer to refer to the full Introduction and Discussion:

“...human nonverbal vocalisations, which can resemble those of nonhuman mammals (e.g., [12]), are often characterised by extreme fluctuations in voice pitch and, like those of other mammals, their acoustic structure typically maps onto their purported biological or social function [13]. This form-function mapping across vocal types appears to function to exploit listeners’ perceptual sensitivities and biases. For example, distress vocalisations (cries, screams) are typically several orders of magnitude louder and higher-pitched than modal speech, attracting attention and eliciting arousal [14]. Such distress vocalisations also exhibit different spectral profiles than those of agonistic or aggressive vocalisations (roars, grunts), which are typically characterised by a relatively low F_0 and a high proportion of nonlinear phenomena (e.g., [15,16]).

Given this considerable variability, an important question is whether acoustic cues to vocaliser attributes are shared across call types (e.g., as observed in red deer: [17] or are call-specific (e.g., as observed in zebra finches: [18]). Indeed, in humans, no previous study has examined whether individual differences in F_0 are preserved across the broad range of human vocal sounds, from modal and emotionally valenced speech to nonverbal vocalisations. While it is possible that idiosyncratic differences in F_0 between different vocalisers might be overridden by the more extreme F_0 modulations that characterise agonistic and distress vocalisations, existing data on the F_0 profiles of human grunts [15], roars [16], laughs and cries [19] indicate that such vocalisations retain a degree of sexual dimorphism, wherein men produce relatively lower-pitched vocalisations than do women. There is also preliminary evidence that within each sex, F_0 in modal speech correlates with F_0 in sung speech [20], and that cues to individual identity are retained in valenced human speech [21], laughter [22], cries [23], and in the screams of both humans [24, cf. 25] and nonhuman primates [26] (with the caveat that speaker recognition is substantially reduced from these vocalisations compared to modal speech among human listeners [21,22,25]). Finally, longitudinal studies have recently revealed that individual differences in F_0 remain relatively stable across the lifespan, from infancy to childhood and throughout adulthood [27,28]. Taken together, this body of literature suggests that F_0 may represent a reliable and surprisingly stable individual marker despite its extreme dynamicity at the within-vocaliser level (e.g., across vocal types). From an

evolutionary and ethological perspective, such *F0* stability could function to provide honest information about a vocaliser’s identity and biosocial profile (e.g., dominance, masculinity) regardless of the mode of vocal production, or social context.”
(Lines 78 - 112)

Another way in which I think the authors oversell their case is by omitting discussion of the fact that *F0* also varies within and between individuals not just for biological reasons, but also due to learned and social factors. This is particularly important here due to the use of actors from a single training tradition, rather than naturally occurring utterances or at least acted performances drawn from two cultures with different emotion display rules. This point requires some discussion and should be mentioned as a significant limitation of the study.

We agree and have further expanded on the introductory text to emphasize that *F0* can also vary due to social factors.

“In addition to fluctuating dynamically during speech production within a single vocaliser [3,4 for reviews], *F0* varies widely across individuals, both between the sexes (men’s voices are lower-pitched than women’s) and within adults of the same sex, due to a combination of endocrinological, physiological, anatomical, and social factors [5].” (Lines 54 – 58)

“Humans can readily modulate their voice *F0* to express a range of emotions and motivations during speech production [3,4], and studies conducted in real-life contexts have shown that men and women alter their speech *F0* depending on social context or whom they are speaking to (see e.g., [8,9]).” (Lines 67 – 70)

We also emphasize the need for further work on cross-cultural samples.

“...given the influence of social and cultural factors on voice production and vocal expressions of emotion, additional research is also needed to examine the generalisability of these results to other cultures.” (Lines 297 – 299)

Although we had already noted in our Discussion that future studies should examine both volitional (acted) and spontaneous (naturally occurring) vocalisations, we now suggest specifically examining naturally occurring vocal displays produced in real-life social contexts:

“As researchers have observed some differences between simulated and spontaneous vocal sounds (e.g., identity-related information is more reliably encoded in genuine laughter [19,22]), future studies may also compare *F0* relationships in volitional versus spontaneous vocal types, including naturally occurring vocal displays produced in real-life social contexts.” (Lines 304 – 308)

Finally, while our results suggest that individual differences in F_0 are likely to be preserved across social contexts (as they are preserved across highly diverse vocal types), we note that this should be explicitly tested in future work.

“Thus, our results also suggest that individual differences in F_0 may be preserved across social (and emotional) contexts, though this hypothesis should be explicitly tested.” (Lines 311 – 313)

I also don't completely agree that the acoustic space in speech is constrained by rules of language. If this is true, then why study speech during (simulated) pain or fear or aggression, as they do here? What about singing, which is linguistic, but particularly in modern vocalization styles is quite unconstrained in terms of voice quality? At a minimum, this claim requires citation and justification

Emotional speech and singing contain both verbal and nonverbal elements. We agree with the reviewer that these vocal types utilise a broader acoustic space than modal speech, including a higher proportion of nonlinear phenomena and broader F_0 range, as our results also support (fig 2). However, where intelligibility in phonetic information is to be retained, emotional speech and singing remain more constrained than vocalisations. Yes - speech can be whispered, screamed or sang, but when it is, intelligibility is largely reduced. Similarly, sung speech with extremely high/low F_0 or harsh, chaotic elements is difficult to understand.

As suggested, we have expanded this statement and added citations and examples to support it:

“Such vocalisations almost certainly emerged in the human vocal repertoire before words [3], and can exploit a broader, less constrained acoustic space than modal speech. Indeed, intelligible speech production necessitates a relatively steady F_0 ensured by an ability to maintain constant subglottal pressure [2], as well as a dense harmonic structure (thus relatively low F_0) to effectively encode phonetic information, including discriminable formant frequencies [11].” (Lines 73 – 78)

Similarly, the conclusion that human voice pitch functions as a stable individual marker due to anatomical and mechanistic constraints is simplistic, given the huge range of variability an individual can produce (as their own data show).

As described in the Discussion, our results and those of several previous studies showing evidence for individual recognition across vocal types [ref 16-20] and stability in F_0 across the lifespan [22-23] suggest that human F_0 is a stable individual marker, *despite its extreme dynamicity* (p 13).

We now clarify that:

“With regards to voice F_0 , this may be due to anatomical constraints (likely linked to individual differences in vocal fold length) that maintain

the relative interindividual differences in *F0* across the typical ranges for each vocal type.” (Lines 282 – 285)

F0 ranges almost completely overlap for males and females, for every utterance type except modal speech, further undermining this argument.

Our data do not support this. While men’s *F0*’s fall within the lower *F0* range of women’s, women’s *F0* maxima far exceed those of men for all vocal types (fig 2). We have also conducted additional analyses on the recommendation of Reviewer 3 that further show significant sex differences in *F0* for each vocal type (see Results and table S2).

Finally, in their interpretation of the data the authors seem to imply that all statistically reliable correlation values are somehow equivalent, although there’s a big difference between 13% and 64% variance accounted for. Conclusions are not equally strong in all cases here, particularly given the differences between males and females in patterns of association. Again, I’m sure there’s something going on here, but it seems to me that results are subtler and weaker than the authors state.

We agree and have now reported confidence intervals for all correlations (Table S3) and have commented on the potential ecological relevance of relatively weak correlations ($r = 0.10 - 0.30$) in our Discussion:

“Nevertheless, given that these relationships explain less than 10% of the variance in shared *F0* across vocal types, their ecological relevance is uncertain.” (Lines 292 – 294).

Specific quibbles:

“Hertz” should be capitalized throughout the manuscript.

Hertz is now capitalised throughout.

Scientific writing is precise. No perceptual assessment was undertaken, so the use of “pitch” in this paper is inappropriate. Replace with “*F0*” throughout.

We have replaced “pitch” with “*F0*” except where referring to the perception of *F0* e.g., in reference to the results of previous playback studies, listeners’ judgments.

The word “despite” at the beginning of the introduction implies that use of voice for speech is somehow in conflict with non-linguistic uses of phonation, which isn’t really true. I suggest rewriting this sentence (possibly by just deleting the first clause).

We have changed “despite” to “in addition to”:

“In addition to being the carrier of language, the human voice has been shaped by selection to communicate biologically relevant traits of the vocaliser [1 for review].” (Lines 48 – 49)

Line 145: errant apostrophe in ERBs.

“ERB’s” has been changed to “ERBs” throughout the manuscript.

Reviewer: 3

This study examined whether inter-individual differences in fundamental frequency (F0) are correlated across vocal types. The authors reported novel findings that human F0 are preserved across vocal types (valenced speech and vocalizations), suggesting that F0 might be a reliable biosocial marker across communication contexts. The research question is interesting, and findings are relevant to recent reports in literature, which examine the perceptual effect of human non-verbal vocalizations (screams, roars, etc.) that are beyond modal speech. Nevertheless, there are some concerns with the study’s sample size, and these limitations should be highlighted in discussion.

We have taken the helpful Reviewer’s comments into account during the revision of the manuscript, as described below in bold. Thank you for your time and suggestions.

Major Concerns:

-One of the concerns with the study is its sample size ($n = 23$ for women and $n = 28$ for men). The authors, for example, concluded that “...modal speech was a poor predictor of the pitch of both valenced speech and nonverbal vocalizations (p.12, line 220).” I conducted a quick G-power post-hoc analysis with one-tailed test using women ($n = 23$), spearman’s $\rho = .09$ (between modal speech and aggressive speech), and p value of .05, which produces the statistical power 0.11. Hence, the probability of avoiding a Type II error is low.

We agree and have now reported confidence intervals for all correlations (see Table S3). We also emphasise this limitation in our Discussion, while remarking on the potential ecological relevance of weak correlations:

**“The results of the present study warrant replication on a larger, more representative sample, as the current study may be unpowered to detect statistical significance in weak correlations ($r = 0.10 - 0.30$). Nevertheless, given that these relationships explain less than 10% of the variance in shared F0 across vocal types, their ecological relevance is uncertain.”
(Lines 290 – 294)**

-Some of the voice recordings were collected from drama or acting students. Livingstone et al. (2014) reported that vocal training and acting experiences influence vocalizer’s pitch and other measures. <https://www.ncbi.nlm.nih.gov/pmc/articles/PMC3945712/> Perhaps, the correlation in inter-individual differences in pitch across vocal types could be driven

by the unique characteristics of the sample. Please consider reporting the number of drama students in the sample. Are there differences in F0 variations between acting students' and University of Sussex students'?

Participants were recruited either from the Royal Central School of Speech and Drama (London, UK) or the University of Sussex, but all were acting or drama students.

We have included the following text in our Discussion to address the Reviewer's point, with citation to papers showing evidence both for and against actor-effects on vocal production:

“Replication studies should also include individuals without acting experience, as the potential influence of vocal training on the spectral and perceptual dimensions of speech and nonverbal vocalisations remains unclear (see e.g., [36,37]).” (Lines 294 – 297)

“... future studies may also compare F0 relationships in volitional versus spontaneous vocal types, including naturally occurring vocal displays produced in real-life social contexts.” (Lines 306 – 308)

Minor Points

-Consider running independent t-tests or 2-way mixed ANOVAs (Sex x Vocal Type), with reports on Means and SDs, to report the degree of sexual dimorphism in F0 across vocal type. Judging from the figure, it appears F0 dimorphism is greatest in aggressive roar and fear scream.

We thank the Reviewer for this excellent recommendation. Our manuscript now includes Results of Linear Mixed Models followed by pairwise comparisons (Šidák corrected) showing significant sex differences in F0 for each vocal type. To emphasize the magnitude of sexual dimorphism in each vocal type we have included both Hz and ERB values for the male/female violin plots in figure 2 as well as asterisks to indicate significant sex differences (all $p < .001$).

-I suggest some modifications for Figure 1. What do the numbers on box plot represent? Are they IDs of the outlier, or F0 value in ERB? Also consider changing F to Female and M to Male. If not, including notes in the figure caption would clarify the points.

The Reviewer is referring to Figure 2 (violin plots). As recommended, we have created a new legend on the plot in which F and M are now labelled Female and Male, respectively. We have also expanded the figure caption to more clearly identify the values (F0 means) given above each violin plot, which now include both Hz and ERB values:

Figure 2 [caption]. “Violin plots representing the full distribution in mean F0 for each sex and each vocal type. Mean F0 values are given in both

Hertz and ERBs above the violin plot for each sex and each vocal type. Significant sex differences were observed for each vocal type (*) $p < .001$ following Šidák correction). Plots were produced in R ggplot 2 package.”**

-It's great to see that Spearman's rho for all analyses (rs), both significant and non-significant, were reported in Table 1. Nevertheless, plotting these rank relationships in x-y plot would provide additional information. Since Spearman analyses only consider monotonic relationships, curvilinear relationships, for example, may not be detected without visualization.

Another excellent suggestion, thank you. We have now included scatterplot matrices for all pairwise comparisons as supplementary figure S1. Significant correlations are indicated with shading. As the Reviewer can see, significant correlations were positive and linear, and there is no clear evidence of non-linear relationships for nonsignificant comparisons.

-Are there similar lines of evidence in nonhuman animals, in which inter-individual differences in fundamental frequency (F0) that are correlated across vocal types? It would be a nice discussion point.

We strongly agree that this is an interesting point, and have noted this in our Introduction and Discussion sections:

“Given this considerable variability, an important question is whether acoustic cues to vocaliser attributes are shared across call types (e.g., as observed in red deer: [17] or are call-specific (e.g., as observed in zebra finches: [18]).” (Lines 89 – 91)

“While few studies have investigated the stability of F0 differences across the vocal repertoires of other species, there is evidence that cues to individual identity may also be preserved across call types in nonhumans mammals, including red deer [17], rhesus macaques [34] (Rendall et al. 1996), and cows [35].” (Lines 285 – 289)

Appendix B

Response to Reviewers

RSOS-191642-R2

Reviewer: 1

Comments to the Author(s)

I review the revised manuscript “Individual differences in human voice pitch are preserved from speech to screams, roars and pain cries” (RSOS-191642.R1).

While most comments (including those of the other reviewers) have been very well addressed, I must say that one major weakness of this study, the sample size (as pointed out by me and R3), has not been well addressed in the manuscript.

A sensitivity analysis was not provided and the current discussion of this clear limitation (“The results of the present study warrant replication on a larger, more representative sample, as the current study may be unpowered to detect statistical significance in weak correlations ($r = 0.10 - 0.30$). Nevertheless, given that these relationships explain less than 10% of the variance in shared F0 across vocal types, their ecological relevance is uncertain.”) seems to miss the point. First, as pointed out in my previous review, the study is even only moderately powered for $r_s > .3$. For example, with $N = 23$ you have power $= .62$ ($\alpha = .05$, one tailed) to detect an effect of $r = .4$ (which is typically considered a medium to large effect size). Thus, the study is, in general, clearly underpowered. Second, even if the observed estimate is high (e.g., $r = .55$) looking at the 95% CIs tells us that this is a really imprecise estimate which might, in well-powered studies, fall far below this estimate (e.g., $r = .20$).

In conclusion, I would hope that the authors can discuss this limitation in another revision. Further, I see no reason for putting the 95% CIs in the supplementary table only. It should be possible to fit those in Table 1.

We thank the Reviewer for their time and final comments. As suggested, we have now further expanded our discussion of the limited sample size and thus lack of power in our correlational analyses. We have included the 95% CI's directly in Table 1 and explicitly refer readers to these ranges, underscoring that confidence intervals are broad for most correlations. We have also removed the final sentence “Nevertheless, given that...” as we agree with the Reviewer that this sentiment, while true, distracted from the main point we were making here – that the study warrants replication on a larger and more representative sample to ensure the robustness and replicability of the effects. (please see Line 293 in the ‘clean’ version of the revised ms).

Reviewer: 2

Comments to the Author(s)

Review of RSOS-191642-R1

Pisanski et al., Individual differences in human voice pitch are preserved from speech to screams, roars, and pain cries

The authors have done an excellent job of revising this interesting paper, and have addressed all my concerns to my satisfaction. It remains only for me to quibble about apostrophe use:

Line 164: bandwidth's ◊ bandwidths

Lines 207, 208, 209: ERB's ◊ ERBs

Lines 213, 238, 261, 266: F0's ◊ F0s

Line 252: men ◊ men's

We thank the Reviewer for this positive feedback and for the grammatical edits, which we have incorporated in the final revision of the manuscript.

Reviewer: 3

Comments to the Author(s)

The authors satisfactorily addressed my concerns, including other reviewers', with the manuscript. I do not have any additional comments.

We thank the Reviewer for their time.